# Essential Thrombocythemia: One-Center Data in a Changing Disease

**DOI:** 10.3390/medicina58121798

**Published:** 2022-12-06

**Authors:** Nicoleta Pirciulescu, Mihnea-Alexandru Gaman, Marina Mihailescu, Cristina Constantin, Mihaela Dragomir, Camelia Dobrea, Simona Costache, Iulia Ursuleac, Daniel Coriu, Ana Manuela Crisan

**Affiliations:** 1Department of Clinical Hematology, Center of Hematology and Bone Marrow Transplantation, Fundeni Clinical Institute, 022328 Bucharest, Romania; 2Faculty of Medicine, “Carol Davila” University of Medicine and Pharmacy, 050474 Bucharest, Romania; 3Department of Molecular Biology, Center of Hematology and Bone Marrow Transplantation, Fundeni Clinical Institute, 022328 Bucharest, Romania; 4Department of Hematopathology, Center of Hematology and Bone Marrow Transplantation, Fundeni Clinical Institute, 022328 Bucharest, Romania; 5Department of Cytomorphology, Center of Hematology and Bone Marrow Transplantation, Fundeni Clinical Institute, 022328 Bucharest, Romania

**Keywords:** essential thrombocythemia, *JAK2*, IPSET score, thrombosis, survival

## Abstract

*Introduction:* Essential thrombocythemia is a chronic myeloproliferative neoplasm associated with thrombo-hemorrhagic events and the progression to myelofibrosis or acute myeloid leukemia. The purpose of this article is to present real-world data on ET cases diagnosed and managed between 1998 and 2020 in the largest, tertiary hematology reference center in Romania and to evaluate the impact of thrombotic events on survival. *Methods:* A real-world, retrospective cohort-type study was conducted. We collected and statistically analyzed data from 168 patients who met the 2016 WHO diagnostic criteria for ET and who were managed between 1998 and 2020 in our center. *Results:* The median age at diagnosis of ET was 51.8 years, with a female predominance (66.07%). The *JAK2V617F* mutation was detected in 60.71% of patients. Leukocytosis at diagnosis was associated with a higher risk of thrombosis, and *JAK2V617F*-positive cases exhibited a 1.5-fold higher risk of developing thrombotic events. The average survival in ET with major thrombosis was 14.5 years versus 20.6 years in ET cases without major thrombosis. Other predictors of survival were high-risk IPSET score and age >60 years. *Conclusions:* Romanian patients diagnosed with ET are generally younger than 60 years and are predominantly female. The occurrence of thrombotic events was influenced by gender, leukocyte count at diagnosis and *JAK2V617F* positivity. Survival was impacted by age, the presence of *JAK2V617F* mutation, hypertension, major thrombotic complications and IPSET score. Notably, these findings warrant careful interpretation and further confirmation in the setting of prospective studies.

## 1. Introduction

Essential thrombocythemia (ET) is a classical chronic myeloproliferative neoplasm (MPN) which, according to the 2016 World Health Organization (WHO) classification [1,2], is associated with a good prognosis and a median survival in patients aged less than 60 years at diagnosis of 33 years. In less than a half of ET subjects, the most frequent symptoms are asthenia, weight loss, fever and early satiety. The onset and/or progression of ET may be complicated by thrombotic events and/or the progression to myelofibrosis and/or acute myeloid leukemia (AML). A recent meta-analysis estimated the prevalence of thrombotic events in ET at 20.7% (16.6–25.5). The most common arterial events reported in ET are stroke (10.7%) and acute coronary syndromes (6.1%), whereas the most frequent venous events are deep vein (3.4%), pulmonary (0.9%), splanchnic (1.4%) and cerebral (0.7%) thrombosis, respectively. The prevalence of hemorrhagic events in ET is evaluated at 7.3% (5.3–10.0), the most common being cutaneous/mucosal and/or gastrointestinal bleedings [3]. The risk of progression to myelofibrosis at 15 years varies between 4 and 11%, and that to AML varies between 2 and 5% [4]. Another meta-analysis pointed out that the distribution of somatic mutations in ET is as follows: *JAK2* (31.3–72%), *CALR* (12.6–50%) and *MPL* (0.9–12.5%) gene mutations, respectively [5].

The purpose of this article is to present real-world data on ET cases diagnosed and managed between 1998 and 2020 in the largest tertiary hematology reference center in Romania, namely, the Center of Hematology and Bone Marrow Transplantation of the Fundeni Clinical Institute, with a particular focus on the occurrence of thrombotic events and their impact on overall survival in ET. In our view, this is the first study to report the real-world characteristics of ET patients from our country.

## 2. Materials and Methods

The present article reports the results of a cohort-type retrospective study which included a total of 168 patients hospitalized between 1998 and 2020 in the Center of Hematology and Bone Marrow Transplantation, Fundeni Clinical Institute, Bucharest, Romania and who met the WHO 2016 criteria for ET [2]. For the subjects who presented to our hematology department before 2016, we applied the criteria retrospectively to ensure that they were correctly classified as ET. All pathology slides were reviewed by an experienced hematopathologist.

For this retrospective chart review, we applied the following inclusion criteria: patients aged ≥18 years who met the WHO 2016 diagnostic criteria for ET and who had available electronic and/or printed records. The exclusion criteria were represented by: children (age <18 years), diagnosis of myeloproliferative neoplasm other than ET (chronic myeloid leukemia, polycythemia vera, primary myelofibrosis etc.), secondary/reactive thrombocytosis and cases miscoded as ET in the electronic records of the hospital. The study was conducted following the Declaration of Helsinki and was approved by the Ethics Committee of Fundeni Clinical Institute (approval number 46897 on 29/08/2022). Informed consent was sought from all patients to allow their printed/electronic records and their medical data to be used for research purposes.

Briefly, for the detection of mutations in the *JAK2* and *CALR* genes, DNA was isolated from peripheral blood granulocytes using a commercially available DNA extraction kit (PureLink Genomic DNA, Thermo Fisher Scientific, Waltham, MA, USA). *JAK2V617F* was detected by an Amplification-Refractory Mutation System Polymerase Chain Reaction (ARMS-PCR) assay using a previously described method [6]. CALR exon 9 type I (deletion of 52 pb–c.1092_1143del) and type II (insertion of 5 bp–c.1154_1155ins) mutations were detected by multiplex PCR using a method previously described by Jeong et al. [7].

Data analysis: The collected data were entered in tables, and the statistical analysis was computed using Epi Info 7.0 and SPSS 16.0. Demographic data, clinical characteristics and laboratory results were analyzed by descriptive analysis. The comparison of continuous variables was performed by independent *t*-tests, and that of categorical variables was performed by chi-squared tests. To evaluate predictors of mortality, we used ROC curves. The significance of differences was determined if the *p*-value was less than 0.05. For survival analysis, we used Kaplan–Meier curves. To compare survival curves, we used the log-rank test. Graphical representations of the results were obtained using SPSS 16.0 and Microsoft Office Excel.

## 3. Results

### General Characteristics of ET Patients

A total of 500 patients were classified as having ET in the electronic records system of our hospital. After applying the inclusion and exclusion criteria, we identified 168 subjects who were diagnosed and managed in the Center of Hematology and Bone Marrow Transplantation of the Fundeni Clinical Institute, Bucharest, Romania during 1998–2020 and who met the WHO 2016 diagnostic criteria for ET. The mean age at diagnosis was 51.80 years (range: 20–84 years), and the distribution based on sex showed a female dominance (66.07% women). Genetic testing identified somatic mutations in 66.08% of the study group, with *JAK2V617F* being the most commonly detected mutation (60.71%), followed by mutations in the *CALR* (4.76%) and *MPL* (0.60%) genes. One patient had triple-negative ET, 26.68% of the subjects were *JAK2V617F*-negative and 11 patients (6.54%) did not display mutations in the *JAK2* or *CALR* genes (Table 1).

The most frequent symptoms at diagnosis were: asthenia (16.67%), headache (13.09%) and abdominal discomfort (11.90%). The most common comorbidities in our ET cohort were hypertension (34.13%), heart diseases (22.62%), dyslipidemia (20.36%), liver disease (14.97%), gastrointestinal disorders (13.77%), smoking (12.50%) and type 2 diabetes mellitus (8.98%). According to the IPSET score, the subjects were stratified as high-risk (47.03%), low-risk (26.79%) and intermediate-risk (26.19%). In terms of treatment options, 23.21% (n = 39) of the study group received antiplatelet agents or anticoagulants alone, 26.19% (n = 44) of ET cases were prescribed only cytoreduction therapy and 47.02% (n = 79) of the subjects were given both cytoreductive agents and antiplatelets or anticoagulants. Of the patients who received cytoreduction alone, 28 subjects were prescribed hydroxycarbamide (hydroxyurea, HU), 14 patients received anagrelide and 2 patients received interferon alpha-2b (IFN). Of the patients receiving cytoreductive treatment plus antiplatelets or anticoagulants, 53.17% (n = 42) received HU and aspirin, 17.72% (n = 14) received HU and anticoagulants (ACO), 24.05% (n = 19) received anagrelide and aspirin, 2.53% (n = 2) received anagrelide and anticoagulants (ACO) and 2.53% (n = 2) received IFN and aspirin. These findings are depicted in detail in Table 1.

## 4. Complications

A total of 51 (30.35%) patients experienced thrombotic events. Three patients developed two thrombotic complications. In terms of timing, 22.22% of the thrombotic events occurred before the ET diagnosis, 64.81% occurred at the time of diagnosis and 12.97% occurred after the diagnosis at variable intervals. Thrombotic events were arterial (60%) or venous (34%), and 6% of subjects developed both arterial and venous events. The most frequent arterial events were stroke (27.78%), myocardial infarction (9.25%) and peripheral arterial disease (5.56%), whereas the most common venous event was thrombosis in the splanchnic territory (22.22%) (Table 2). The mean age of ET patients who developed stroke was 58.42 ± 13.70 years. All stroke subjects had cardiovascular risk factors, and in 46.67% of them, the thrombotic complication occurred before the diagnosis of ET, with a latent period varying from 1 to 5 years. All cases of myocardial infarction occurred as inaugural events in the diagnosis of ET, and the mean age of the patients who were diagnosed with ET following this complication was 52.20 ± 12.67 years. Similarly, thrombotic events in the splanchnic territory were inaugural events in the diagnosis of ET. The mean age of the subjects who experienced these complications was 41.58 years, and the sex ratio (male:female) was 1:1.

We recorded a total of four hemorrhagic events: two cases of subarachnoid hemorrhage before the diagnosis of ET and two cases of gastrointestinal bleeding which occurred after the diagnosis of ET but while the subjects were on antiplatelet agents. One patient underwent liver transplantation due to chronic liver failure secondary to Budd–Chiari syndrome.

The progression to myelofibrosis was observed in four patients, as confirmed by bone marrow trephine biopsy specimens which were reviewed by an experienced hematopathologist. The time to progression varied from 1 to 16 years. No progression to AML was recorded.

### 4.1. Comparison between ET Patients with/without Major Thrombosis

In this study, 40 (23.81%) patients developed major thrombotic events: 11 patients experienced symptoms associated with thrombotic complications (focal neurologic deficits, dyspnea and angina pectoris), 19 patients reported headaches and 17 patients suffered from asthenia (Table 3). Female sex emerged as a protective factor for thrombosis (RR = 0.46, 95% CI (0.27–0.79), *p* = 0.008). There was a tendency for smoking in ET subjects who developed thrombotic complications (20% versus 10.16%, *p* > 0.05). In terms of laboratory parameters, individuals with ET who experienced major thrombotic events were more likely to present with leukocytosis at diagnosis (*p* = 0.01). In our study, 70% of the patients who developed major thrombosis were *JAK2V617F*-positive and exhibited a 1.5-fold higher risk of developing a thrombotic event (RR = 1.50, 95% CI (0.82–2.75); *p* = 0.23). In our investigation, the IPSET score was higher in the ET-with-major-thrombosis subgroup (87.5% versus 34.37%) (Table 3).

In 19.61% (n = 10) of ET patients with major thrombosis, hereditary thrombophilia mutations were detected: 50% (n = 5) of the patients had mutations in the MTHFR gene, 20% (n = 2) of patients had mutations in the PAI-1 gene, 10% (n = 1) of patients had factor V Leiden thrombophilia, 10% (n = 1) of patients had factor II thrombophilia and 10% (n = 1) of patients had factor XIII thrombophilia.

### 4.2. Survival Analysis

In our study, the general mortality was 13.07% (n = 22/168). Mortality was higher in the ET-with-major-thrombosis subgroup, i.e., 25% (n = 10/40) versus 9.38% (n = 12/128). Subjects who developed major thrombotic events had a 2.5 times higher risk of mortality (RR = 2.66; 95% CI 1.24–5.70, *p* = 0.02).

Age was a predictor of mortality in the general ET group (AUC 0.793 CI 95% (0.693–0.893); *p* = 0.0001) but also in the ET-with-major-thrombosis group (AUC 0.715 95% CI (0.528–0.902), *p* = 0.04).

In our study, gender did not influence mortality in ET (RR = 0.89 95% CI 0.40–2.01; *p* = 0.98).

In the ET-with-major-thrombosis group, 60% of the patients who died during the study had hypertension at the time of diagnosis, having a three-times-higher risk of death (RR-3.11 95% CI (1.05–9.15); *p* = 0.05). *JAK2V617F* mutation tended to be associated with a higher risk of death in the general group, but its impact on mortality was not statistically significant (RR = 2.2, 95% CI (0.85–5.67); *p* = 0.13). However, 77.27% of the subjects who died during the study were *JAK2V617F-*positive.

The long-term survival analysis showed a statistically significant shorter survival of patients with ET and major thrombosis. The average survival in this subgroup was 14.5 years (95% CI (12.2–16.9)) versus 20.6 years (95% CI (19.3–21.9)) in the ET subgroup without major thrombosis (*p* = 0.007).

The average long-term survival tended to beshorterin*JAK2V617F-*positive ET cases, i.e., 14.9 years (95% CI (13.5–16.3)) versus 21.3 years (95% CI (19.9–22.7)) in *JAK2V617F-*negative ET individuals (*p* = 0.06). The average survival according to the IPSET score was notably lower in high-risk patients (Table 4).

Survival was shorter in *JAK2V617F-*positive subjects aged ≥60 years as compared to *JAK2V617F-*negative individuals aged <60 years (Table 4).

None of the laboratory parameters were predictors of mortality: hemoglobin (AUC 0.462, 95%CI (0.293–0.631) *p* = 0.60), platelets (AUC 0.564, 95%CI (0.412–0.716), *p* = 0.38), leukocytes (AUC 0.620, 95%CI (0.462–0.778), *p* = 0.08), PDW (AUC 0.554, 95%CI (0.418–0.689), *p* = 0.43), MPV (AUC 0.580, 95%CI (0.437–0.723), *p* = 0.24), LDH (AUC 0.498, 95%CI (0.333–0.663), *p* = 0.97). Survival was only influenced by age and JAK2V617F status (Figure 1).

## 5. Discussion

ET is a chronic myeloproliferative neoplasm characterized by a variable clinical picture, thrombo-hemorrhagic complications and, sometimes, an unpredictable evolution. The aim of our study was to analyze the characteristics and survival of patients with ET from the largest hematology reference center in Romania.

In our assessment, the average age at ET diagnosis was 51.8 years. Our patients tended to be younger, as it has been reported in the literature that ET is generally diagnosed at the age of 60 years or later [8,9]. We detected a higher prevalence of ET in women and demonstrated that the female sex was a protective factor for thrombosis. This finding is in line with other investigations in which the male sex was a risk factor for venous thrombotic events (RR 1.99 (1.03–3.83); *p* = 0.03) [8,9,10].

In our investigation, leukocytosis at diagnosis (*p* = 0.01) was associated with a higher risk of thrombotic complications. Other authors have confirmed that ET subjects with white blood cell counts higher than 11 × 10^9^/mmc (classified as high-risk) have almost a doubled risk of arterial thrombosis (RR 1.76 (1.05–2.97), *p* = 0.03) (HR 1.66 (1.01–2.72), *p* = 0.04) [10].

According to our findings, ET individuals who harbor the *JAK2V617F* mutation had a 1.5-fold higher risk for thrombotic events. This is in accordance with previously published data which depicted a doubled risk of thrombosis among *JAK2V617F*-positive ET subjects (OR 1.92, 95% CI 1.45–2.53), irrespective of whether the thrombotic complications were arterial (OR 1.77, 95% CI 1.29–2.43) or venous (OR 2.49, 95% CI 1.71–3.61) in nature [11].

The prevalence of major thrombotic events and of splanchnic thrombosis was in consonance with other investigations [10,11,12].

The overall mortality was 13.17%, which is in agreement with previous reports which delineate a lower average survival in men diagnosed with ET (14 years versus 20 years) [13,14,15]. In our investigation, ET patients who were associated with hypertension had a three-fold higher risk of mortality (RR-3.11 95% CI (1.05–9.15); *p* = 0.05). According to the literature, subjects suffering from ET and cardiovascular disease at diagnosis have a doubled risk of mortality [16].

Our findings point out that long-term survival was lower in the ET-with-major-thrombosis subgroup (14.5 versus 20.6 years) (*p* = 0.007). This result has also been confirmed by another study that analyzed predictors of survival in 244 ET patients. In the aforementioned assessment, individuals with ET who developed arterial thrombotic events during follow-up had a survival of 97 months versus 140 months (*p* < 0.001) for ET patients without thrombotic events [16]. Long-term survival analysis according to the IPSET score showed a significantly shorter survival in high-risk (14 years) versus intermediate-risk (17 years) and low-risk patients (22 years). The impact of the IPSET score on survival was validated in three other cohort studies that included subjects from Germany, France and Italy, respectively. In Germany, the average survival in the low-risk category was not reached (95% CI: 15.6-NR); however, survival was 17.5 years (95% CI: 13.6-NR) in the intermediate-risk group and 9.4 years (95% CI: 5.8–13.4) in the high-risk group, respectively [17]. In France, the average survival according to the IPSET score was 18.6 years (95% CI: NR) in the low-risk category and 7.1 years (95% CI: 5.8–13.4) in the high-risk category [18]. In Italy, the average survival was not reached for the low-risk group, but it was 24.5 years (95% CI 22.3-NR) in the intermediate-risk group and 14.7 years (95% CI 11.9–18) in the high-risk group [18]. Long-term survival assessment based on *JAK2V617F* status and age demonstrated that survival is lowest in *JAK2V617F-*positive ET patients aged ≥60 years. Many studies have identified an age >60 years as one of the factors associated with lower survival in ET alongside anemia, male sex and a history of thrombosis [13]. Regarding the impact of somatic mutations on ET survival, previous investigations have not highlighted a statistically significant difference between *JAK2V617F*-positive and *JAK2V617F*-negative cases, regardless of age [19].

Our study has several strengths and some limitations. A major strength is that we presented real-world data on ET cases diagnosed and managed between 1998 and 2020 in the largest tertiary hematology reference center in Romania, namely, the Center of Hematology and Bone Marrow Transplantation of the Fundeni Clinical Institute, with a particular focus on the occurrence of thrombotic events and their impact on overall survival in ET. In our view, this is the first study to report real-world characteristics of ET patients from our country. Moreover, we ensured that all cases met the WHO 2016 criteria, including via the revision of all pathology slides by an experienced hematopathologist.

Nevertheless, we must underline that the retrospective nature of our investigation was associated with several caveats. Firstly, the study sample was relatively small, even if ET is a rare disease, which limits the extrapolation of the results. Secondly, our research was based on the observational and retrospective assessment of medical records, and data related to the comorbidities or complications encountered in our ET cohort might have been omitted from their files, as the electronic records of our hospital were not designed to collect data for research purposes. In addition, the analysis was based on data collected from a single reference center, which may reflect the generalizability of the findings. Another limitation of our research is that we were unable to assess the impact of treatment and other factors on the patients’ evolution, e.g., MPL gene mutations, and, thus, selection or information biases and/or confounding factors remain. In addition, as compared to a prospective study, our retrospective analysis provides an inferior level of evidence and only association, but not causation, could be demonstrated. We need further studies based on larger patient samples and multicentric data with a prospective design to have a better understanding of ET and its management in our country.

## 6. Conclusions

Romanian patients diagnosed with ET are generally younger than 60 years and are predominantly female. The occurrence of thrombotic events was influenced by gender, leukocyte count at diagnosis and *JAK2V617F* positivity. Survival was impacted by age, the presence of *JAK2V617F* mutation, hypertension, major thrombotic complications and IPSET score. Notably, these findings warrant careful interpretation and further confirmation in the setting of prospective studies.

## Figures and Tables

**Figure 1 medicina-58-01798-f001:**
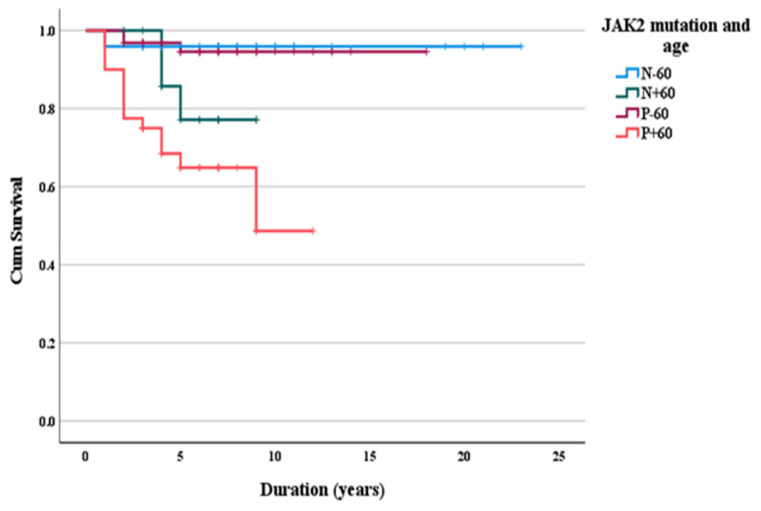
Long-term survival according to age and *JAK2V617F* status.

**Table 1 medicina-58-01798-t001:** General characteristics of ET patients.

Demographic Data	N = 168 Patients
Mean age	51.80 ± 15.63 years
Female sex	111 (66.07%)
**Mutations**	
*JAK2 V617F*	102 (60.71%)
*CALR* (type I: 4 cases; type II: 4 cases)	8 (4.76%)
*JAK2 V617F*-negative *JAK2V617F*-negative, *CALR*-negative *JAK2 V617F*-negative, *CALR*-negative, *MPL*-negative	45 (26.78%) 11 (6.54%) 1 (0.6%)
**Symptoms/signs at diagnosis**	
Asymptomatic	87 (51.78%)
Asthenia	28 (16.67%)
Headache	22 (13.09%)
Abdominal discomfort	12 (7.14%)
Paresthesias	15 (8.92%)
Symptoms associated with MI, PE, stroke (angina, dyspnea, focal neurologic deficits)	11 (6.54%)
Weight loss	9 (5.35%)
Bone pain	4 (2.38%)
Concentration problems	4 (2.38%)
Splenomegaly	7 (4.16%)
Erythromelalgia	3 (1.78%)
Hepatomegaly	5 (2.97%)
**Comorbidities**	
None	42 (25%)
Hypertension	58 (34.52%)
Heart diseases	38 (22.62%)
Dyslipidemia	34 (20.36%)
Liver disease	25 (14.97%)
Type 2 diabetes mellitus	15 (8.98%)
Gastrointestinal disorders	23 (13.77%)
COPD	4 (2.40%)
CKD	6 (3.59%)
Other neoplasia in medical history	2 (1.20%)
Others (endocrine or psychiatric disorders)	17 (10.11%)
Smoking	21 (12.50%)
**Laboratory data**	
Mean platelet count (M/μL)	886.30 ± 380.50
Median white blood cell count (M/μL)	9.58 [8.22–11.97]
Mean hemoglobin level (mg/dL)	14.14 ± 2.13
Mean hematocrit value (%)	42.39 ± 6.02
Mean MPV value (fL)	9.99 ± 0.77
Mean PDW value (fL)	11.72 ± 1.68
Median LDH level (U/L)	250.50 [194–336]
Long spleen axis (cm)	11.64 ± 1.81
**IPSET score**	
Low-risk	45 (26.79%)
Intermediate-risk	44 (26.19%)
High-risk	79 (47.03%)
**Treatment**	
Antiplatelets/Anticoagulants	39 (23.21%)
Aspirin	37/39
Anticoagulants	1/39
Aspirin + anticoagulants	1/39
Cytoreductive treatment	44 (26.19%)
HU	28/44
Anagrelide	14/44
IFN	2/44
Cytoreductive treatment + antiplatelets/anticoagulants	79 (47.02%)
HU + aspirin	42/79
HU + anticoagulant	14/79
Anagrelide + aspirin	19/79
Anagrelide + anticoagulants	2/79
IFN + aspirin	2/79
Cytoreductive treatment + aspirin + anticoagulants	6 (3.57%)

Legend: CALR, calreticulin gene. CKD, chronic kidney disease. COPD, chronic obstructive pulmonary disease. dL, deciliter. HU, hydroxyurea/hydroxycarbamide. IFN, interferon alpha-2b. IPSET, International Prognostic Score for Essential Thrombocythemia. JAK2, Janus Kinase 2 gene. L, liter. LDH, lactate dehydrogenase. MI, myocardial infarction. MPL, myeloproliferative leukemia virus oncogene. MPV, mean platelet volume. mg, milligrams. PDW, platelet distribution weight. PE, pulmonary embolism. U, units.

**Table 2 medicina-58-01798-t002:** Thrombotic events.

Type of Thrombotic Event	N = 54 Events
**Arterial**	
Stroke	15 (27.78%)
TIA	2 (3.70%)
MI	5 (9.25%)
Peripheral arterial disease	3 (5.56%)
Others (retinal thrombosis, erythromelalgia)	9 (16.67%)
**Venous**	
Splanchnic	12 (22.22%)
DVT	3 (5.56%)
PE	3 (5.56%)
Others (thrombophlebitis)	2 (3.70%)
**Time of onset**	
Thrombotic events before ET diagnosis	12 (22.22%)
Thrombotic events at ET diagnosis	35 (64.81%)
Thrombotic events after ET diagnosis	7 (12.97%)
Major thrombotic events	43 (79.63%)
Minor thrombotic events	11 (20.37%)
**Age of onset**	
Major thrombosis < 60 years	24 (55.81%)
Major thrombosis > 60 years	19 (44.19%)

Legend: DVT, deep vein thrombosis. ET, essential thrombocythemia. IPSET, International Prognostic Score for Essential Thrombocythemia. MI, myocardial infarction. PE, pulmonary embolism. TIA, transient ischemic attack.

**Table 3 medicina-58-01798-t003:** Comparison between patients with ET with/without thrombotic events.

Demographic Data	ET with Major Thrombotic Events (N = 40)	ET without Thrombotic Events (N = 128)	*p*-Value
Mean age (years)	52.30 ± 16.72	51.64 ± 15.34	0.81
Male sex	36 (28.13%)	21 (52.50%)	**0.008**
**Clinical features**			
Asymptomatic	7 (17.5%)	80 (62.50%)	**<0.001**
Asthenia	11 (27.50%)	17(13.28%)	0.06
Abdominal discomfort	12 (30.00%)	8 (6.25%)	**0.001**
Paresthesia	3 (7.50%)	12 (9.38%)	1.00
Weight loss	3 (7.50%)	6 (4.69%)	-
Headache	3 (7.50%)	19 (14.84%)	-
Symptoms associated with MI, stroke, PE (angina, dyspnea, focal neurologic deficits)	11 (27.5%)	-	-
Splenomegaly	4 (10.00%)	3 (2.34%)	-
Hepatomegaly	3 (7.50%)	2 (1.56%)	-
**Comorbidities**			
Heart disease	10 (25.00%)	28 (21.88%)	0.84
Hypertension	13 (32.50%)	45 (35.16%)	0.90
Dyslipidemia	11 (27.50%)	23 (17.97%)	0.27
Type 2 diabetes mellitus	4 (10.00%)	11 (8.59%)	1.00
Liver disease	11 (27.50%)	14 (10.94%)	**0.02**
Smoking	8 (20.00%)	13 (10.16%)	0.17
None	4 (10.00%)	38 (29.69%)	**0.02**
**Laboratory data**			
Mean platelet count (M/μL)	887.44 ± 374.68 × 10^9^	885.96 ± 383.67 × 10^9^	0.98
Median white cell count (M/μL)	11.32 [8.76–14.41] × 10^9^	9.50 [7.97–11.25] × 10^9^	**0.01**
Mean hemoglobin level (g/dL)	13.94 ± 2.26	14.20 ± 2.09	0.52
Mean value of hematocrit (%)	41.33 ± 6.40	42.70 ± 5.90	0.23
Mean value of PDW (fL)	12.27 ± 2.22	11.55 ± 1.45	0.15
Mean value of MPV (fL)	10.20 ± 0.98	9.93 ± 0.69	0.16
Median LDH level (U/L)	278 [203–431]	235 [191–331]	0.15
**Mutations**			
*JAK2V617F-*positive	28 (70.00%)	74 (57.81%)	0.23
*CALR*	0 (0.00%)	8 (6.25%)	-
*MPL*	0 (0.00%)	1 (0.78%)	-
*JAK2V617F*-negative	12 (30.00%)	45 (35.16%)	0.54
**IPSET score**			
Low	2 (5.00%)	43 (33.60%)	
Intermediate	3 (7.50%)	41 (32.03%)	
High	35 (87.50%)	44 (34.37%)	
**Type of treatment**			
Antiplatelets/Anticoagulants	3 (7.50%)	36 (28.13%)	
Cytoreduction	5 (12.50%)	39 (30.47%)	
Cytoreduction + Antiplatelets/Anticoagulants	26 (65.00%)	53 (41.40%)	
Cytoreduction + Antiplatelets + Anticoagulants	6 (15.00%)	0 (0.00%)	

Legend: CALR, calreticulin gene. ET, essential thrombocythemia. IPSET, International Prognostic Score for Essential Thrombocythemia. JAK2, Janus Kinase 2 gene. L, liter. LDH, lactate dehydrogenase. MI, myocardial infarction. MPL, myeloproliferative leukemia virus oncogene. MPV, mean platelet volume. mg, milligrams. PDW, platelet distribution weight. PE, pulmonary embolism. U, units. TIA, transient ischemic attack.

**Table 4 medicina-58-01798-t004:** Long-term survival according to age and *JAK2V617F* status.

	Median Survival (Years)	95% CI	*p*-Value
Age ≥60 years and *JAK2V617F-*positive	8.16	6.56–9.76	
Age ≥60 years and *JAK2V617F-*negative	7.94	6.88–9.00	
Age <60 years and *JAK2V617F-*positive	17.19	16.31–18.08	
Age <60 years and *JAK2V617F-*negative	20.10	20.88–23.32	<0.001

## Data Availability

Not applicable.

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
