# Peer review of "Essential Thrombocythemia: One-Center Data in a Changing Disease"

_medicina, 2022, doi:10.3390/medicina58121798_

Round 1
Reviewer 1 Report
The article presents the data collected from patients with ET over about twenty years in a clinical institute; the focus is on thrombotic events and their impact on overall survival.
The paper is a retrospective evaluation of patient’s data and does not assess the effect of treatment. Unfortunately, it does not give new or particular news on the topic.
The number of patients evaluated is moderate, while larger cohort are available in the literature.
Major issues:
line 61: If in the evaluated period 500 ET patients were studied and only in 168 of them had analyzable data, only about 1/3 of the patients have been considered. This represent a significant bias. Why other patients were not evaluable?
lines 67-68: If 33.9% of patients were Jak2 negative and 4.76% are JAK2 and CALR negative but only 0.6% are 3NEG it seems that 4.1% of patients are MPL mutated, while in the table 1, MPL mutated patients are 0.6%. The Authors must clarify and better explain this sentence and correlate it with the data in the table
line 70: abdominal discomfort is typically present in children with ET, but uncommon in adults. Does this be related to splenomegaly? It appears unlikely being splenomegaly present in only 4% of patients. This point has to be clarified
line 90: Thrombotic events after diagnosis seem to be low for the use of appropriate treatment. The lack of correlation of events with adopted therapy is crucial
line 92-93: arterial thrombotic events were very common in this cohort. A correlation with patients’ age and the disease duration is necessary to better evaluate this data
line 101: Chronic liver failure usually is due not to portal thrombosis that produces a portal cavernoma, but to Budd Chiari Syndrome.
Lines 102-103: Are the Authors sure that the patients evolved into MF were ET and not PMF-0? Data regarding bone marrow biopsy are necessary to clarify
Lines 110-112: while leukocytosis is well known to be associated with increased risk of thrombosis, as the Authors stated in the discussion, longitudinal spleen axis resulted longer in patients with major thrombosis. I suggest to cautiously evaluate this data in comparison to splanchnic thrombotic events.
Lines 123-130: I suggest to report all the cause of mortality, in particular considering that age is and important predictor (not surprisingly)
Table 1:
- weight loss and bone pain are typically associated with MF, but extremely rare in ET. Did the Authors re-evaluate these patients’ BOM?
Figures
-Figure 1, 2 and 3 are redundant and can be eliminated together with table 4
Finally, the language is poor and some English usage errors are found.
Author Response
Dear Academic Editor,
Dear Peer-Reviewers,
We are very thankful to you and to the peer-reviewers for the pertinent notes; we have carefully read the comments and have revised/completed the manuscript accordingly. Our responses are given in a point-by-point manner below. All the changes to the manuscript are highlighted in yellow.
We hope that, in this new form, the manuscript will be suitable for publication in Medicina.
Reviewer 1
Response: We would like to thank you for your valuable comments which helped us improve the manuscript. All suggestions were taken into consideration and appropriate information, as well as required corrections, were provided. New/corrected parts are highlighted inyellow to facilitate the assessment of changes. We did our best to fulfil the expectations and we hope that you will be satisfied with our corrections.
All in all, we thank you for your constructive comments regarding our manuscript.
The article presents the data collected from patients with ET over about twenty years in a clinical institute; the focus is on thrombotic events and their impact on overall survival.
The paper is a retrospective evaluation of patient’s data and does not assess the effect of treatment. Unfortunately, it does not give new or particular news on the topic.
The number of patients evaluated is moderate, while larger cohort are available in the literature.
Response: Thank you for your comment. The purpose of this article is to present the real-world data on ET cases diagnosed and managed between 1998 and 2020 in the largest, tertiary hematology reference center, namely the Center of Hematology and Bone Marrow Transplantation of the Fundeni Clinical Institute, with a particular focus on the occurrence of thrombotic events and their impact on overall survival in ET. To our view, this is the first study to report real-world characteristics of ET patients from our country.
Major issues:
line 61: If in the evaluated period 500 ET patients were studied and only in 168 of them had analyzable data, only about 1/3 of the patients have been considered. This represent a significant bias. Why other patients were not evaluable?
Response: Thank you for your comment. We have clarified this misunderstanding in the revised paper. In fact, 500 cases were coded as ET. However, many of these 500 cases were miscoding errors, e.g., secondary thrombocytosis coded as ET at the first presentation of the patient, primary myelofibrosis coded as ET until the results of the bone marrow trephine biopsy came in, etc. Some of the cases did not have electronic or printed records available or simply did not meet the WHO 2016 diagnostic criteria for ET. It is important to clarify that all pathology slides of reviewed by an experienced hematopathologist to ensure that all the 168 met the criteria for ET and that no other case from the 500 was excluded from the analysis if it met the WHO 2016 diagnostic criteria for ET.
lines 67-68: If 33.9% of patients were Jak2 negative and 4.76% are JAK2 and CALR negative but only 0.6% are 3NEG it seems that 4.1% of patients are MPL mutated, while in the table 1, MPL mutated patients are 0.6%. The Authors must clarify and better explain this sentence and correlate it with the data in the table
Response: Thank you for your valuable comment. At present, we only test for JAK2V617F and CALR gene mutations. Patients with MPL gene mutations performed the genetic test in private molecular biology labs, however, most of our patients cannot afford these costs. We must point out that we are one of the few hematology centers in our country that carries out these tests for free because we have our own Molecular Biology Lab with experienced staff. However, ET diagnosis can be established based on the first 3 major criteria + 1 minor criterion even if we cannot ascertain MPL gene mutations in JAK2-negative and CALR-negative cases. In the near future we hope to be able to clarify the genetic status of all ET cases and implement MPL gene mutation testing in our own lab.
line 70: abdominal discomfort is typically present in children with ET, but uncommon in adults. Does this be related to splenomegaly? It appears unlikely being splenomegaly present in only 4% of patients. This point has to be clarified
Response: Thank you for your valuable comment.Abdominal pain canbe related to splenomegaly or hepatomegaly (7.13%) or it can also be caused by some gastrointestinal disorders encountered in our patients (e.g., irritable bowel syndrome).
ine 90: Thrombotic events after diagnosis seem to be low for the use of appropriate treatment. The lack of correlation of events with adopted therapy is crucial
Response: Thank you for your valuable comment.
line 92-93: arterial thrombotic events were very common in this cohort. A correlation with patients’ age and the disease duration is necessary to better evaluate this data
Response: Thank you for your valuable suggestions. We have performed the suggested analysis.
line 101: Chronic liver failure usually is due not to portal thrombosis that produces a portal cavernoma, but to Budd Chiari Syndrome.
Response: Thank you for your valuable comment. Indeed, we checked the patient’s electronic records and chronic liver failure was due to Budd-Chiari syndrome.
Lines 102-103: Are the Authors sure that the patients evolved into MF were ET and not PMF-0? Data regarding bone marrow biopsy are necessary to clarify.
Response: Thank you for your valuable suggestions.Yes, we are sure that the patients evolved into MF. We performed bone marrowtrephine biopsies both at the diagnosis of ET and when the evolution to MF occurred and the pathology slides were reviewed by the same experienced hematopathologist with more than 25 years of expertise in this field. We are fortunate enough to benefit from the expertise of a hematopathologist who is a member of the author team of this publication (together with our experienced colleagues from the Molecular Biology and Cytomorphology Lab).
Lines 110-112: while leukocytosis is well known to be associated with increased risk of thrombosis, as the Authors stated in the discussion, longitudinal spleen axis resulted longer in patients with major thrombosis. I suggest to cautiously evaluate this data in comparison to splanchnic thrombotic events.
Response: Thank you for your valuable suggestions.Unfortunately, noultrasound examination of the spleen was performed before the occurrence of thrombotic events in the splanchnic territory.
Lines 123-130: I suggest to report all the cause of mortality, in particular considering that age is and important predictor (not surprisingly).
Response: Thank you for your valuable suggestions.Unfortunately, out of the 22 deceased patients, only 2 patients died in the hospital. The death of the other 20 patients was confirmed by the health insurance system, but we do not know the exact cause of death. In their case, we considered the date of death as their last presentation to the hospital as they were patients who attended the outpatient department monthly for clinical examination, blood tests and HU/anagrelide prescriptions.
Table 1:
- weight loss and bone pain are typically associated with MF, but extremely rare in ET. Did the Authors re-evaluate these patients’ BOM?
Response: Thank you for your comment. As previously mentioned, the same pathologist reviewed the slides of all these cases. Our pathologist had been working in the hematopathology department of our center for more than 20 years.
Figures
-Figure 1, 2 and 3 are redundant and can be eliminated together with table 4
Response: Thank you for your valuable suggestions. We have eliminated these figures.
Finally, the language is poor and some English usage errors are found.
Response: Thank you for your valuable suggestion. The scientific writing and English language of the paper was reviewed by a native speaker of English certified by the University of Cambridge, UK. In addition, if accepted, minor errors will be fixed during proofreading.
Sincerely yours,
The authors

Reviewer 2 Report
The quality of presentation is high, the results are in general similar to literature and represents one-center experience.In my oppinion the general conclusions must be added at the end of the article
Author Response
Dear Academic Editor,
Dear Peer-Reviewers,
We are very thankful to you and to the peer-reviewers for the pertinent notes; we have carefully read the comments and have revised/completed the manuscript accordingly. Our responses are given in a point-by-point manner below. All the changes to the manuscript are highlighted in yellow.
We hope that, in this new form, the manuscript will be suitable for publication in Medicina.
Reviewer 2
The quality of presentation is high, the results are in general similar to literature and represents one-center experience.In my oppinion the general conclusions must be added at the end of the article
Response: We would like to thank you for your valuable comments which helped us improve the manuscript. All suggestions were taken into consideration and appropriate information, as well as required corrections, were provided. New/corrected parts are highlighted inyellow to facilitate the assessment of changes. We did our best to fulfil the expectations and we hope that you will be satisfied with our corrections.
General conclusions have been added to the manuscript.
Thank you for your valuable suggestions. We do hope you will find the revised version of the paper significantly improved and worthy of publication in Medicina.
Sincerely yours,
The authors
